# Influence of Frailty Syndrome on Kinesiophobia According to the Gender of Patients after Coronary Artery Bypass Surgery

**DOI:** 10.3390/healthcare9060730

**Published:** 2021-06-14

**Authors:** Martyna Kluszczyńska, Agnieszka Młynarska, Wioletta Mikulakova

**Affiliations:** 1Department of Gerontology and Geriatric Nursing, Faculty of Health Sciences, Medical University of Silesia, 40-635 Katowice, Poland; amlynarska@sum.edu.pl; 2Department of Physiotherapy, Faculty of Health Care, University of Presov, 080 01 Presov, Slovakia; wioletta.mikulakova@unipo.sk

**Keywords:** coronary artery bypass, frailty syndrome, kinesiophobia

## Abstract

(1) Background: Kinesiophobia is associated with fear of movement, general fitness exercises, and physical or mental discomfort. In patients with frailty syndrome, in addition to coexisting conditions, the postoperative recovery period may be longer than in patients without frailty; (2) Methods: The study included 108 people over 60 years of age, qualified for CABG (coronary artery bypass grafting). The Tilburg Frailty Index was used to assess frailty syndrome and the kinesiophobia scale was to assess fear of physical activity; (3) Results: Frailty syndrome was diagnosed among 19.44% of respondents. The social components of frailty were more intense in the group of women *p* = 0.009. The mean results for the biological and psychological domain on the scale of kinesiophobia were 1.94 and 1.6. The level of kinesiophobia was higher among women than among men taking into account the psychological domain (*p* = 0.006) and the subdomains: motor skills self-assessment (*p* = 0.042) and body care (*p* = 0.011); (4) Conclusions: Frailty syndrome does not affect kinesiophobia among patients after CABG. The level of kinesiophobia was higher among women than among men, taking into account the psychological domain. The greater the energy resources, the lower the level of frailty and its physical components in the group of women.

## 1. Introduction

Frailty syndrome (FS) is defined as a physiological syndrome, which is characterized by a decrease in the reserves of resistance to stressors, resulting from the accumulation of reduced capacity of different physiological systems, leading to adverse consequences. It is characterized by a reduced ability to maintain and rebuild the homeostasis of the organism and it is presented as an entity other than disability or coexisting disease. Frailty syndrome is not synonymous with old age and multi-disease, but all dysfunctions of the body are a risk factor for the syndrome, and disability is a negative effect [1,2,3].

Frailty syndrome in Poland is diagnosed in 6.7% of elderly people, including more than 30% of patients aged 75–80 years and 50% over 80 years of age. Patients with cardiovascular problems present symptoms of frailty syndrome three times more often than others. The coexisting conditions such as atherosclerosis, diabetes mellitus, and hypertension additionally influence the multitude of complications and symptoms before and after surgery to which patients are subjected [4].

The kinesiophobia occurring in patients after cardiac surgery is associated with fear of movement, of general fitness exercises, as well as physical or mental discomfort. According to the literature, kinesiophobia is a manifestation of personality predisposition to physical activity. It occurs most often as a consequence of injury in the form of irrational fear of all forms of activation, it is associated with anxiety and psychosomatic symptoms such as excessive sweating or tremors of hands. Kinesophobia can be considered in two domains—biological, which includes anxiety of pain, fatigue, exhaustion, and psychological, which includes the fear of ridicule due to lack of physical activity. Both domains influence each other and form a complementary picture of the patient with kinesophobia. Sociodemographic factors such as gender can influence the level of fear of movement [5,6].

Postoperative rehabilitation plays an important role in preventing complications after cardiac surgery. Lack of participation or avoidance of activity may negatively affect the return to preoperative form as well as self-service capacity. Early activation of the patient prevents lower limb thrombosis, allows for free flow of postoperative drainage, and facilitates expectoration of the secretion in the bronchial tree affecting proper breathing. Avoiding complications should be a priority in the care of the elderly. Early identification of risk groups and implementation of appropriate measures to minimize the possibility of adverse events, improve the quality of life of patients and interrupts the cascade of frailty syndrome [7].

Cardiovascular diseases are one of the main problems among elderly patients. Thanks to the development of medicine, patients may undergo various cardiological and cardiac surgeries, often without being aware of the high perioperative risk. An important element in the diagnosis and qualification of patients for surgery is the balance between the benefits of the procedure performed and the occurrence of complications. Early diagnosis of FS in a group of patients makes it possible to refer them to specialist clinics and prepare relatives for care. However, there is a difficulty in unambiguously identifying the frailty syndrome due to the occurrence of other accompanying diseases and numerous symptoms [8].

The aim of the study was to assess the influence of frailty syndrome on kinesiophobia after coronary artery bypass surgery according to the gender of patients. 

## 2. Materials and Methods

### 2.1. Study Design and Settings

The study was conducted in the Cardiac Surgery Department in Katowice among the participating patients. All respondents were qualified for coronary artery bypass surgery. It was an observational, consecutive enrollment, and cross-sectional study.

### 2.2. Study Participants and Selection

The patients included in the study were over 60 years old, were qualified for coronary artery bypass surgery, and agreed to participate in the study, and their mental state enabled them to understand the questions included in the questionnaire. Respondents who were simultaneously qualified for another procedure, e.g., valve replacement, and refused to participate in the follow-up visit were excluded.

### 2.3. Stages of the Study

The study was conducted in two stages. In the first one, the prevalence of the frailty syndrome among patients was assessed, patients were subjected to standardized questionnaires, clinical history and demographic data were collected, and anthropometric measurements were made. In the second stage, after 6 months ± 2 weeks from the procedure, a follow-up visit including a clinical history was made and questionnaires were completed once again.

### 2.4. Ethical Considerations

The study was approved by the Bioethical Committee of the Medical University of Silesia in Katowice, KNW/0022/KB/22518. Before joining the study, participants were informed about the confidentiality of the study, anonymity, goals, and methodology. Patients were also informed about the possibility to withdraw from each stage of the study. The data collection and analysis were conducted on the basis of ethical principles, which were derived from the Helsinki Declaration.

### 2.5. Research Instruments

The following standardized tools were used in this work.

The Tilburg Frailty Indicator (TFI) was used to evaluate the frailty syndrome. The frailty syndrome was evaluated taking into account both physical as well as mental and social dysfunctions. The first part of the scale concerned sociodemographic information such as age, gender, marital status, education level, country of origin, and the second part consisted of 15 questions relating to the occurrence of the main components of frailty. It covered 3 domains: physical, psychological, and social. The total score could be between 0 and 15 points. Frailty syndrome is recognized by a TFI score ≥ 5. The tool was developed by Gobbens et al. and is based on the concept of the frailty model [9,10].

Fear of physical activity was assessed using the kinesiophobia scale. The questionnaire allowed the assessment of dysfunctions in two domains—biological and psychological and 7 subdomains. The first one was an average of morphological parameters, stimulation demand, energy resources, and biological drives, and the second one was self-acceptance, self-assessment of motor skills, and body care. The questionnaire contained 20 questions describing different behaviors that were evidence of kinesiophobia. Respondents evaluated them on a scale of 1–5, where higher numbers meant that they perceived given behaviors more strongly. The result for each of the subdomains was the average number of points from the questions included in a given subdomain. Therefore, the results could be interpreted according to the key for the single question in which:-“1” meant “I completely disagree”, and therefore indicated a complete lack of kinesiophobia,-“2” meant “I partially disagree”, and thus indicated a lack of kinesiophobia rather than its presence,-“3” meant “I don’t know, I don’t have an opinion”, and so it was indicative of an intermediate state: neither of the presence of kinesiophobia nor of its absence,-“4” meant “I agree,” and thus testified to the presence of kinesiophobia rather than the absence thereof,-“5” meant “I completely agree” and therefore showed a strong kinesiophobia [5].

### 2.6. Statistical Methods

The comparison of the values of qualitative (not numerical) variables in the group of women and men was made with the use of the chi-square test (with Yates correction for 2 × 2 tables) or the Fisher’s exact test where low expected numbers appeared in the tables.

Comparison of the values of quantitative (numerical) variables in the group of women and men was performed using the Mann-Whitney test.

Correlations between quantitative variables were analyzed with the Spearman correlation coefficient. There were no missing data.

In the analysis, a significance level of 0.05 was assumed. Thus, all *p* values below 0.05 were interpreted as indicating significant correlations. The analysis was performed in the R program, version 4.0.2 [11].

The Strengthening and reporting of observational studies in epidemiology (STROBE) statement reporting guidelines were followed in this study.

## 3. Results

The group consisted of 108 people—69.4% men and 30.5% women. The mean age of the respondents was 70.06 ± 6.36 years, including the group of women in the range of 67–76 years, and men 64–74.5 years. Body height and weight were significantly greater among men, *p* = 0.001. Most of the respondents (40.74%) were overweight, and obesity concerned 20.37% of the respondents (based on BMI). More than half of the patients (56.48%) showed no limitation in physical activity, excessive fatigue, or dyspnea according to the New York Heart Association (NYHA) class. Women were widowed more often than men and less often lived with their spouse or partner *p* < 0.001. Detailed data are presented in the Table 1. 

Frailty syndrome was diagnosed among 19.44% of respondents, including 27.27% of women and 16% of men. The social components of frailty were significantly more intense in the group of women *p* = 0.009. In the remaining domains, no statistically significant differences were observed. Details are presented in the Table 2.

Mean results for the biological and psychological domain on the scale of kinesiophobia in the studied group were 1.94 and 1.61, which rather indicated the lack of limitations of motor activity after surgery (mean rounded points equaled 2). Similar results were observed in the particular subdomains.

The level of kinesiophobia was significantly higher among women than among men taking into account the psychological domain (*p* = 0.006) and the subdomains: self-assessment of motor skills (*p* = 0.042) and body care (*p* = 0.011). In women, the most points were awarded to the level of energy resources (1.89) and the least to morphological parameters (1.79). In the male group, the results showed a total lack of kinesiophobia in the body care subdomain (mean score of 1.45), while the highest score was given to morphological parameters (2.05). Detailed data are presented in the Table 3.

Comparing the components of frailty syndrome and the domain of kinesiophobia for the whole study group, no significant relationships were observed (all *p* > 0.05). After the adjustment of the alpha value of the *p*-value was applied, none of the relationships between the TFI subscales and kinesiophobia were significant anymore, so it may have been a false-positive result. The data are presented in Table 4.

In the group of women, the overall result of the TFI and its physical components correlated significantly (*p* ˂ 0.05) and negatively (r ˂ 0) with the level of energy resources, i.e., the higher the energy resources, the lower the level of frailty and its physical components and vice versa: the higher the level of frailty and its physical components, the lower the energy resources. Details are presented in Table 5.

In the group of men, similarly as in relation to the whole, there were no significant relationships (all *p* > 0.05). Details are presented in Table 6.

## 4. Discussion

The problem of reduced physical activity after surgical procedures occurs in an increasing number of patients. It is becoming the cause of deteration of the quality of life and also has a negative impact on the return to independence. Dąbek et al. have determined the degree of kinesiophobia in people with coronary artery disease and the relationship between physical activity and socio-demographic conditions and morbidity. Based on the group of 217 patients, whose average age was 67 years, they came to the following conclusions. Over 70% of hospitalized patients showed a high degree of kinesiophobia (based on the Tampa Scale > 37 points). Cardiovascular diseases were the main cause of limitations of motor activity. No effect of gender, BMI, demographic variables, or disease duration on the Tampa Scale results was observed. In the original studies, no high level of kinesiophobia was observed—this may be due to the fact that the study was conducted 6 months after surgery, where the level of kinesiophobia may have decreased. The influence of gender on kinesiophobia was also observed in contrast to the studies of Dąbek [12]. Other conclusions were drawn by Kocjan and Knapik. Based on the study group n = 115 people, they determined the influence of determinants such as age, gender, and health condition on the level of kinesiophobia in patients undergoing rehabilitation. The patients showed an increased level of fear of movement and these were more often women. The psychological domain had more influence on kinesiophobia than the biological domain. The same conditions were observed in our own research. It turned out that subjective assessment of the patient’s health condition is associated with increased anxiety. The worse the self-assessment, the higher the level of kinesiophobia. Anxiety may cause worsening of cardiological rehabilitation results. The aim of comprehensive cardiovascular rehabilitation is to eliminate cardiovascular risk factors and restore full physical activity. The authors’ own research also showed a relationship between kinesiophobia and the gender of the patients. Women presented a higher level of anxiety in the psychological domain [13].

Nascimento et al. assessed the impact of pain on physical activity anxiety and quality of life. Analyzing the data, it turned out that kinesiophobia may be an unfavorable factor in the deterioration of quality of life and functional capacity [14].

The fear of physical activity was also felt by patients after myocardial infarction. The authors Åhlund et al. examined the influence of cardiological rehabilitation on the level of kinesiophobia among 62 patients over 60 years old. They observed statistically significant correlations between the alleviation of anxiety and fear of physical activity and exercise with a physiotherapist. Avoidance of anxiety occurred among 48% of all respondents, including 62% of the rehabilitated and 29% of the control group. After 4 months, there were no statistically significant differences between the groups. Physical activity increased over time since myocardial infarction. In our research, all patients underwent a 4-week cardiac rehabilitation, which could significantly reduce the limitations of motor activity, which was also observed by Ahlund researchers and co-authors [15].

On the basis of 332 respondents with diagnosed coronary artery disease, Back et al. came to the following conclusions. Kinesophobia was diagnosed in 20% of the respondents. Participation in cardiac rehabilitation, a high level of physical activity, and good general health conditions reduced the chance of kinesiophobia among the respondents. Heart failure and the occurrence of hospital complications significantly increased the risk of movement anxiety. It turned out that ischemic heart disease is associated with high levels of kinesiophobia; therefore, in secondary prevention among this group of patients, it is important to implement increased cardiac rehabilitation. Patients after aorto-coronary bypass surgery were rehabilitated as quickly as possible, which is why in my research the level of kinesiophobia was low [16].

The research of Silva et al., which was based on a group of patients aged 79.4 ± 7.03 years, made the following conclusions. The occurrence of kinesiophobia was closely related to the age of the respondents. Pain-induced disability increased with age and was associated with the risk of falls and frailty. In our research, the influence of the frailty syndrome on kinesiophobia was not noticed after the coronary artery bypass surgery, perhaps after the expansion of the research group due to the higher age (the average age in our research was 70.06 years), the frailty syndrome would occur in more patients [17].

Knapik et al. investigated the level of kinesiophobia among patients with coronary artery disease. The research group consisted of 135 patients over 60 years of age. It turned out that over 76 percent of the respondents had a high level of kinesiophobia, and unlike our own research, gender had no effect on the occurrence of anxiety. Education was the only one, among the socio-demographic variables, which differentiated the kinesiophobia. Better educated people had a lower level of the condition [18].

The influence of sociodemographic factors on the occurrence of kinesiophobia was also noted by Morri et al. As in the authors’ own research, female sex may affect a higher level of perioperative anxiety. The researchers also included age and body weight as unfavorable factors influencing kinesiophobia [19].

### Limitations

When analyzing the available literature, there is a limited number of studies on the impact of the frailty syndrome on kinesiophobia. The discussion used available articles covering research groups in which the age was over 60 years. The prevalence of the frailty syndrome increases with the patient’s age, therefore, even if the authors of the papers in the discussion did not study the frailty syndrome, it was assumed that it may occur in these groups. The study sheds new light on the problem of kinesiophobia among patients after coronary bypass surgery at risk of a syndrome of weakness, which has not been compared in other available scientific studies. The diagnosis of kinesiophobia, as well as the fragility syndrome, should be taken into account in subsequent cardiac rehabilitation, contributing to its improvement.

## 5. Conclusions

Frailty syndrome does not affect kinesiophobia among patients after coronary artery bypass surgery. The level of kinesiophobia was significantly higher among women than among men, taking into account the psychological domain and subdomains: self-assessment of motor skills and body care. The group had a low intensity of fear of physical activity. The greater the energy resources, the lower the level of frailty and its physical components in the group of women. However, considering the use of a multiple repetition correction, there is a risk of a false-positive result.

## Figures and Tables

**Table 1 healthcare-09-00730-t001:** Sociodemographic data of patients included in the study.

Parameter		Gender			*p*
		Women (*N* = 33)	Men (*N* = 75)	Total (*N* = 108)	
Age [years]	mean ± SD	71.18 ± 6.01	69.56 ± 6.49	70.06 ± 1.36	*p* = 0.18
	median	72	68	69.5	
	quartiles	67–76	64–74.5	65–75.25	
Height [cm]	mean ± SD	159.67 ± 6.82	172.16 ± 5.86	168.34 ± 8.43	*p* < 0.001
	median	160	172	169.5	
	quartiles	154–164	168–176	164–175.25	
Body mass [kg]	mean ± SD	71.24 ± 10.91	81.4 ± 13.05	78.3 ± 13.25	*p* < 0.001
	median	70	80	77.5	
	quartiles	65–80	71.5–90	68–87	
Marital status	Married/living with partner	18 (54.55%)	67 (89.33%)	85 (78.70%)	*p* < 0.001
	Unmarried	1 (3.03%)	1 (1.33%)	2 (1.85%)	
	Separated/divorced	0 (0.00%)	3 (4.00%)	3 (2.78%)	
	Widow/widower	14 (42.42%)	4 (5.33%)	18 (16.67%)	
Education	No or basic	2 (6.06%)	4 (5.33%)	6 (5.56%)	*p* = 0.737
	Secondary	21 (63.64%)	43 (57.33%)	64 (59.26%)	
	Higher vocational or higher	10 (30.30%)	28 (37.33%)	38 (35.19%)	
Monthly net income in household	901–1200 PLN	0 (0.00%)	2 (2.67%)	2 (1.85%)	*p* = 0.375
	1201–1500 PLN	0 (0.00%)	1 (1.33%)	1 (0.93%)	
	1501–1800 PLN	7 (21.21%)	7 (9.33%)	14 (12.96%)	
	1801–2100 PLN	14 (42.42%)	40 (53.33%)	54 (50.00%)	
	2101 PLN or more	12 (36.36%)	25 (33.33%)	37 (34.26%)	
NYHA class	I	14 (42.42%)	47 (62.67%)	61 (56.48%)	*p* = 0.117
	II	16 (48.48%)	24 (32.00%)	40 (37.04%)	
	III	3 (9.09%)	4 (5.33%)	7 (6.48%)	

Abbreviations: SD = Standard Deviation; PLN = polish currency, NYHA = New York Heart Association classes of heart failure; BMI = Body Mass Index.

**Table 2 healthcare-09-00730-t002:** Comparison of frailty syndrome domains in the group of men and women.

TFI		Group		*p*
		Women (*N* = 33)	Men (*N* = 75)	
Overall TFI	mean ± SD	2.39 ± 1.73	1.97 ± 1.79	*p* = 0.183
	median	2	2	
	quartiles	1–5	1–2.5	
Physical components	mean ± SD	1.42 ± 1.23	1.33 ± 1.33	*p* = 0.609
	median	1	1	
	quartiles	1–2	0–2	
Psychological components	mean ± SD	0.39 ± 0.56	0.37 ± 0.59	*p* = 0.737
	median	0	0	
	quartiles	0–1	0–1	
Social components	mean ± SD	0.58 ± 0.71	0.27 ± 0.53	*p* = 0.009
	median	0	0	
	quartiles	0–1	0–0	

Abbreviations: TFI = Tilburg Frailty Indicator; SD = Standard Deviation.

**Table 3 healthcare-09-00730-t003:** Comparison of domains and subdomains of kinesiophobia among men and women.

The Scale of Kinesiophobia		Group		*p*
		Women (*N* = 33)	Men (*N* = 75)	
Biological domain	mean ± SD	1.86 ± 0.99	1.97 ± 1.04	*p* = 0.549
	median	1.36	1.64	
	quartiles	1.18–2.55	1.14–2.5	
Psychological domain	mean ± SD	1.84 ± 0.9	1.51 ± 0.77	*p* = 0.006
	median	1.44	1.22	
	quartiles	1.22–2.33	1–1.72	
Morphological parameters	mean ± SD	1.79 ± 1.24	2.05 ± 1.44	*p* = 0.829
	median	1.5	1	
	quartiles	1–2	1–3	
Individual demand for stimulation	mean ± SD	1.86 ± 1.13	2.01 ± 1.25	*p* = 0.671
	median	1.67	1.33	
	quartiles	1–2	1–2.67	
Level of energy resources	mean ± SD	1.89 ± 1.11	1.98 ± 1.2	*p* = 0.608
	median	1.25	1.25	
	quartiles	1–3	1–2.75	
The power of biological drives	mean ± SD	1.85 ± 1.19	1.83 ± 1.2	*p* = 0.752
	median	1	1	
	quartiles	1–3	1–2.75	
Level of self-acceptance	mean ± SD	1.8 ± 1.29	1.62 ± 1.05	*p* = 0.563
	median	1	1	
	quartiles	1–2	1–2	
Self-assessment of motor skills	mean ± SD	1.86 ± 1.01	1.52 ± 0.85	*p* = 0.042
	median	1.33	1	
	quartiles	1–2.33	1–1.67	
Body care	mean ± SD	1.85 ± 1.02	1.45 ± 0.8	*p* = 0.011
	median	1.25	1	
	quartiles	1–2.25	1–1.38	

Abbreviations: SD = Standard Deviation.

**Table 4 healthcare-09-00730-t004:** Correlations between the Tilburg Frailty Indicator (TFI) subscale and kinesiophobia.

	Kinesioph- obia	Biological Domain	Psychological Domain	Morphological Parameters	Individual Demand for Stimulation	Level of Energy Resources	Power of Biological Drives	Level of Self-Acceptance	Self-Assessment of Motor Skills	Body Care
TFI										
Overall TFI		r = −0.081, *p* = 0.407	r = −0.051, *p* = 0.599	r = −0.017, *p* = 0.865	r = 0.028, *p* = 0.772	r = −0.084, *p* = 0.387	r = −0.041, *p* = 0.671	r = −0.08, *p* = 0.412	r = −0.053, *p* = 0.586	r = −0.078, *p* = 0.423
Physical components		r = −0.079, *p* = 0.419	r = −0.086, *p* = 0.378	r = −0.041, *p* = 0.671	r = 0.031, *p* = 0.749	r = −0.07, *p* = 0.473	r = −0.034, *p* = 0.724	r = −0.123, *p* = 0.206	r = −0.064, *p* = 0.51	r = −0.09, *p* = 0.352
Psychological components		r = 0.153, *p* = 0.114	r = 0.005, *p* = 0.962	r = 0.166, *p* = 0.086	r = 0.143, *p* = 0.141	r = 0.111, *p* = 0.255	r = 0.167, *p* = 0.084	r = 0.081, *p* = 0.406	r = −0.111, *p* = 0.254	r = −0.032, *p* = 0.746
Social components		r = −0.138, *p* = 0.156	r = 0.091, *p* = 0.349	r = −0.007, *p* = 0.943	r = −0.095, *p* = 0.328	r = −0.121, *p* = 0.211	r = −0.12, *p* = 0.216	r = 0.037, *p* = 0.702	r = 0.14, *p* = 0.148	r = 0.033, *p* = 0.734

Abbreviations: TFI = Tilburg Frailty Indicator; r = correlation coefficient.

**Table 5 healthcare-09-00730-t005:** Correlation between the TFI subscale and kinesiophobia in women.

	Kinesioph- obia	Biological Domain	Psychological Domain	Morphological Parameters	Individual Demand for Stimulation	Level of Energy Resources	Power of Biological Drives	Level of Self-Acceptance	Self-Assessment of Motor Skills	Body Care
TFI										
Overall TFI		r = −0.299, *p* = 0.091	r = −0.007, *p* = 0.969	r = 0.07, *p* = 0.7	r = −0.171, *p* = 0.341	r = −0.389, *p* = 0.025	r = −0.248, *p* = 0.164	r = −0.079, *p* = 0.662	r = −0.148, *p* = 0.411	r = −0.027, *p* = 0.879
Physical components		r = −0.214, *p* = 0.232	r = −0.072, *p* = 0.692	r = 0.177, *p* = 0.324	r = −0.138, *p* = 0.442	r = −0.388, *p* = 0.026	r = −0.167, *p* = 0.353	r = −0.124, *p* = 0.49	r = −0.179, *p* = 0.32	r = −0.044, *p* = 0.81
Psychological components		r = 0.022, *p* = 0.901	r = 0.129, *p* = 0.476	r = 0.111, *p* = 0.538	r = 0.011, *p* = 0.952	r = 0.005, *p* = 0.98	r = 0.059, *p* = 0.745	r = 0.049, *p* = 0.788	r = −0.12, *p* = 0.506	r = 0.077, *p* = 0.669
Social components		r = −0.312, *p* = 0.078	r = −0.08, *p* = 0.657	r = 0.116, *p* = 0.519	r = −0.227, *p* = 0.204	r = −0.283, *p* = 0.11	r = −0.326, *p* = 0.064	r = −0.008, *p* = 0.964	r = 0.002, *p* = 0.99	r = −0.117, *p* = 0.516

Abbreviations: TFI = Tilburg Frailty Indicator; r = correlation coefficient.

**Table 6 healthcare-09-00730-t006:** Correlation between the TFI subscale and kinesiophobia in men.

	Kinesioph- obia	Biological Domain	Psychological Domain	Morphological Parameters	Individual Demand for Stimulation	Level of Energy Resources	Power of Biological Drives	Level of Self-Acceptance	Self-Assessment of Motor Skills	Body Care
TFI										
Overall TFI		r = 0.015, *p* = 0.897	r = −0.122, *p* = 0.298	r = −0.043, *p* = 0.716	r = 0.114, *p* = 0.329	r = 0.044, *p* = 0.71	r = 0.033, *p* = 0.779	r = −0.093, *p* = 0.427	r = −0.062, *p* = 0.599	r = −0.146, *p* = 0.212
Physical components		r = −0.009, *p* = 0.942	r = −0.114, *p* = 0.33	r = −0.103, *p* = 0.378	r = 0.107, *p* = 0.361	r = 0.055, *p* = 0.637	r = 0.015, *p* = 0.899	r = −0.123, *p* = 0.292	r = −0.036, *p* = 0.758	r = −0.117, *p* = 0.318
Psychological components		r = 0.212, *p* = 0.068	r = −0.056, *p* = 0.631	r = 0.188, *p* = 0.106	r = 0.21, *p* = 0.071	r = 0.165, *p* = 0.157	r = 0.213, *p* = 0.067	r = 0.091, *p* = 0.435	r = −0.127, *p* = 0.278	r = −0.096, *p* = 0.412
Social components		r = −0.04, *p* = 0.735	r = 0.092, *p* = 0.433	r = −0.05, *p* = 0.671	r = −0.018, *p* = 0.875	r = −0.026, *p* = 0.823	r = −0.04, *p* = 0.735	r = 0.045, *p* = 0.704	r = 0.157, *p* = 0.178	r = 0.036, *p* = 0.76

## Data Availability

All online surveys are available in the geriatric nursing unit (Katowice, ul. Ziołowa 45/43.

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
