# Peer review of "Influence of Frailty Syndrome on Kinesiophobia According to the Gender of Patients after Coronary Artery Bypass Surgery"

_healthcare, 2021, doi:10.3390/healthcare9060730_

Round 1

Reviewer 1 Report

Manuscript ID: healthcare-1216354
Manuscript title: Influence of frailty syndrome on the kinesiophobia of patients after coronary artery bypass surgery

Major comments
1. Abstract and main text. Revise the following terms: ‘anxiety of movement’ and ‘anxiety of physical activity’ are possibly not synonyms. I suggest using the original terms in your ref. [11], namely ‘fear of movement’ or ‘limitations of motor activity’.
2. Study design, section 2.3 Stages of the study. Being a cross-sectional study (as reported in section 2.1), it is unclear why conducting a follow-up visit for additional data collection with the same questionnaires – particularly as no data about this follow-up is reported.
3. Statistical methods. Why performing a between-sex comparison? There is no indication in the literature review of such hypothesis; even if this is exploratory it should be made explicit.
4. Statistical methods. Considering only the correlation analysis, there are 4*9*3=108 null hypothesis significance tests, posing a high likelihood of false positives. Consider using any method for adjustment of the alpha value of p-value.
5. Results. As per your study aim, tables 1-3 would be more informative if groups were split by frailty rather than sex.
6. Results. Despite the previous comments, following the proposed analysis there should be a ‘Table 6’ with the Correlation between the TFI subscale and kinesiophobia in men.
7. Results. In table 5, the null hypothesis was rejected only in 2/36 tests. In table 4 (and the unshown ‘table 6’) no significant result was found. As no evidence of a Simpson’s paradox appeared when groups were split by sex, these 2 outcomes are likely false-positive test results.
8. Discussion. This section needs extensive revision. There is no discussion about the reported findings in this study. There is no debate between the new findings and those previously reported. There is no information regarding the limitations and strengths of the study, or about the clinical implications of the findings.
9. Conclusions. The final statements regarding the correlation should be revised considering the high likelihood of false-positive test results.

Minor comments
1. Abstract. Define CABG.
2. Methods, Study design. Consider using ‘consecutive enrollment’ where it reads ‘prospective’ to avoid confusion with the cross-sectional study characteristic.
3. Methods, Stage of the study. Revise the ambiguity in the sentence ‘In the first one, all patients were diagnosed 81 with frailty syndrome, were subjected...’.
4. Tables. Replace commas with dots for indicating decimal places.

Author Response

Thanks for the reviews. Below I present the answers to the issues
1. Abstract and main text. Revise the following terms: ‘anxiety of movement’ and ‘anxiety of physical activity’ are possibly not synonyms. I suggest using the original terms in your ref. [11], namely ‘fear of movement’ or ‘limitations of motor activity’. I corrected
2. Study design, section 2.3 Stages of the study. Being a cross-sectional study (as reported in section 2.1), it is unclear why conducting a follow-up visit for additional data collection with the same questionnaires – particularly as no data about this follow-up is reported. The follow-up was to check whether all patients enrolled in the study had coronary bypass surgery, or if another heart defect was diagnosed intra-operatively, which would exclude from the study whether the patients survived the operation. In the research group, all patients underwent bypass surgery and all patients underwent the procedure
3. Statistical methods. Why performing a between-sex comparison? There is no indication in the literature review of such hypothesis; even if this is exploratory it should be made explicit. After reading and analyzing the available and the latest literature (with the consent of the co-authors), I added "by gender" to the title of the work. I also added a paragraph in the introduction about the differences in the sexes of patients, I also changed the purpose of the work and added paragraph in the discussion proving the significant influence of the patients' gender.
4. Statistical methods. Considering only the correlation analysis, there are 4*9*3=108 null hypothesis significance tests, posing a high likelihood of false positives. Consider using any method for adjustment of the alpha value of p-value. I explained this problem in point 7) of the review
5. Results. As per your study aim, tables 1-3 would be more informative if groups were split by frailty rather than sex. I explained this problem in point 3) of the review
6. Results. Despite the previous comments, following the proposed analysis there should be a ‘Table 6’ with the Correlation between the TFI subscale and kinesiophobia in men. I added a table
7. Results. In table 5, the null hypothesis was rejected only in 2/36 tests. In table 4 (and the unshown ‘table 6’) no significant result was found. As no evidence of a Simpson’s paradox appeared when groups were split by sex, these 2 outcomes are likely false-positive test results. After the adjustment of the alpha value of p-value has been applied, none of the relationships between TFI subscales and kinesiophobia is significant anymore, so it may be a false-positive result. I added this theorem in the last paragraph of Statistical Methods8. Discussion. This section needs extensive revision. There is no discussion about the reported findings in this study. There is no debate between the new findings and those previously reported. There is no information regarding the limitations and strengths of the study, or about the clinical implications of the findings. I corrected. I added new paragraphs according to the guidelines and limitations
9. Conclusions. The final statements regarding the correlation should be revised considering the high likelihood of false-positive test results. I explained this problem in point 3) of the review

Minor comments
1. Abstract. Define CABG. I corrected
2. Methods, Study design. Consider using ‘consecutive enrollment’ where it reads ‘prospective’ to avoid confusion with the cross-sectional study characteristic. I corrected
3. Methods, Stage of the study. Revise the ambiguity in the sentence ‘In the first one, all patients were diagnosed 81 with frailty syndrome, were subjected...’. I corrected
4. Tables. Replace commas with dots for indicating decimal places. I corrected

Reviewer 2 Report

Article with well-defined design and very consistent structure. The instruments used seem to be the appropriate ones and revealed that they allow to collect the data that the objectives of the study demanded. In the Introduction, perhaps there was an enrichment of the work, expanding the evidence from other published studies on the themes. The Methodology was clear and consistent. The results and their discussion were well described and evidenced the quality of the dialogue with other studies.

Author Response

Thanks for the reviews      

Round 2

Reviewer 1 Report

Thank you for providing a revised version of your manuscript. Most of my previous comments were properly addressed. There are only three minor issues that could be improved in the manuscript.

1. The new sentence regarding the possible false-positive results after multiple comparison is related to the results and thus would be better addressed in Results rather than Methods section.

2. The Conclusion, particularly, the last sentence, was not revised in light of the highly likely false-positive results as requested.

3. The limitations of the study should be moved to the end of Discussion, and would be better if followed by the strengths of the research as well.

Author Response

Thank you for the new review. All points have been corrected.

Sincerely,

Martyna Kluszczyńka
